# Conceptualising the Integration of Strategies by Clinical Commissioning Groups in England Towards the Antibiotic Prescribing Targets for the Quality Premium Financial Incentive Scheme: A Short Report

**DOI:** 10.3390/antibiotics9020044

**Published:** 2020-01-23

**Authors:** Philip Emeka Anyanwu, Aleksandra J. Borek, Sarah Tonkin-Crine, Elizabeth Beech, Céire Costelloe

**Affiliations:** 1Global Digital Health Unit, Department of Primary Care and Public Health, Imperial College London, London W6 8RP, UK; ceire.costelloe@imperial.ac.uk; 2Nuffield Department of Primary Care Health Sciences, University of Oxford, Radcliffe Observatory Quarter, Oxford OX2 6GG, UK; aleksandra.borek@phc.ox.ac.uk (A.J.B.); sarah.tonkin-crine@phc.ox.ac.uk (S.T.-C.); 3NIHR Health Protection Research Unit in Healthcare Associated Infections and Antimicrobial Resistance, University of Oxford, Oxford OX2 6GG, UK; 4NHS Bath and North East Somerset Clinical Commissioning Group, NHS Improvements London, London BA2 5RP, UK; elizabeth.beech@nhs.net

**Keywords:** primary care, stewardship, antibiotic resistance, AMS campaign, clinical commissioning groups, quality premium

## Abstract

**Background:** In order to tackle the public health threat of antimicrobial resistance, improvement in antibiotic prescribing in primary care was included as one of the priorities of the Quality Premium (QP) financial incentive scheme for Clinical Commissioning Groups (CCGs) in England. This paper briefly reports the outcome of a workshop exploring the experiences of antimicrobial stewardship (AMS) leads within CCGs in selecting and adopting strategies to help achieve the QP antibiotic targets. **Methods:** We conducted a thematic analysis of the notes on discussions and observations from the workshop to identify key themes. **Results:** Practice visits, needs assessment, peer feedback and audits were identified as strategies integrated in increasing engagement with practices towards the QP antibiotic targets. The conceptual model developed by AMS leads demonstrated possible pathways for the impact of the QP on antibiotic prescribing. Participants raised a concern that the constant targeting of high prescribing practices for AMS interventions might lead to disengagement by these practices. Most of the participants suggested that the effect of the QP might be less about the financial incentive and more about having national targets and guidelines that promote antibiotic prudency. **Conclusions:** Our results suggest that national targets, rather than financial incentives are key for engaging stakeholders in quality improvement in antibiotic prescribing.

## 1. Introduction

Clinical Commissioning Groups (CCGs) were established in the English National Health Service (NHS) in April 2013 as the statutory bodies responsible for the planning and commissioning of health care services for their local area [1]. The Quality Premium (QP) is an NHS England initiative introduced in 2013 to reward CCGs financially based on the quality of specific health services considered to be a national or local priority [2]. In order to tackle the public health threat of antimicrobial resistance, improving antibiotic prescribing in primary care practices was included as one of the QP national priorities in the 2015/16 guidance [2] with specific antibiotic targets set each financial year [3]. Although studies have associated recent reductions in antibiotic prescribing in primary care practices in England with the introduction of the QP targets [4,5,6,7], there is still limited understanding of the mechanism of impact of this financial incentive scheme [8].

This paper briefly reports the key findings from a workshop with antimicrobial stewardship (AMS) leads, which aimed to explore the experiences of selecting and adopting strategies within CCGs to help achieve the QP targets on improving antibiotic prescribing. We have also reported a concept mapping activity by the CCGs to develop a conceptual framework for modelling the mechanism of QP effect on antibiotic prescribing.

## 2. Methods

### 2.1. Participant Recruitment

Participants were sent an invitation letter via email with details of the study. We invited 80% of the AMS leads for the 191 CCGs in England (as of May 2019) [1]. Some AMS leads represented more than one CCG [9]. We also invited 12 General Practitioners (GPs) and 3 Nurse Prescribers through the North West London GP Federation.

### 2.2. Group Discussions and Analysis

Participants were assigned to one of three small groups, and each group had a researcher to observe and record discussions. In assigning participants to small working groups, we aimed to have diverse perspectives and experiences in each group in relation to regions and prescribing behaviour of the CCGs represented by the participants. This was important to facilitate comparison of experiences and drive creativity and inclusiveness in the development of the conceptual framework. Discussions within each small group started with participants identifying and summarising the interventions and strategies adopted in their CCGs towards achieving the QP antibiotic prescribing targets. The second part of the workshop was an interactive activity to build a conceptual model to demonstrate the pathways from the QP to antibiotic prescribing through the identified AMS interventions/strategies (potential mediators), and associations between the identified potential mediators, if any.

The modelling activity was followed by a whole-group discussion on key observations about how the QP had been implemented, building on earlier contributions by the participants in the three small groups. This discussion was useful for comparing and summarising the models, looking at triangulation of the data and exploring how the discussed experiences compared between groups and individuals [10].

Detailed notes on discussions and observations within the small and whole groups were taken by PA, AB, CC and other researchers involved in the workshop. The notes were combined and analysed by PA and CC. We conducted a thematic analysis to identify themes across the dataset. The notes were descriptively coded to summarise the key concepts of statements and observations from the group discussions. We reviewed the codes for patterns. Codes that relate to specific fundamental concepts on the experiences of the participants were linked to form themes [11,12]. To improve the reliability of the results, we adopted the Coding Reliability approach recommended by Boyatzis [13]. The developed codes and themes were revalidated and checked for consistency by the researchers who were not part of the initial analysis.

## 3. Results

We recruited 10 AMS leads, three GPs, and one Nurse Prescriber from a diverse range of CCGs and practices in relation to antibiotic prescribing rates and geographical location. Four of the participants who expressed interest did not attend the workshop.

The findings were organised into three main themes, which constituted constructs linking the various aspects of participants’ experiences (discussed below). Figure 1 shows a summarised conceptual model based on the initial models developed by each small group (initial models included as Appendix A).

### 3.1. Strategies Implemented to Help Achieve the QP Antibiotic Targets

Increased CCG engagement with practices on prudent antibiotic prescribing: Practice visits, needs assessments, peer feedback and audits were identified as a set of strategies integrated towards increasing the level of engagement between CCGs and practices, and within practices. Some CCGs used practice visits as an avenue to assess what prescribers need to help them reduce their antibiotic prescribing. Audits to assess antibiotic prescribing in practices against national guidelines were commonly adopted by CCGs alongside feedback to practices to motivate change in prescribing to help meet the QP antibiotic targets. Heavy workload in CCGs and practices was identified as the main barrier to regular antibiotic audit in practices. Participants also reported benchmarking local prescribing data against the national and regional averages as a social norm strategy.

Local financial incentive schemes: The use of other local financial incentives by CCGs to encourage practices to reduce their antibiotic prescribing was widely discussed with most participants recognising the existence of local incentive schemes even before the QP. Some of the local incentives were integrated with the QP targets from 2015.

Other strategies: Other strategies described to help with QP antibiotic targets included the use of AMS training resources—in particular those available on the Treat Antibiotics Responsibly, Guidance, Education, Tools (TARGET) toolkit-, and C-reactive Protein point-of-care testing (CRP POCT).

The participants stressed that the identified strategies were not adopted in isolation. For instance, some of the participants described the integration of audit, feedback, practice visits, and a CCG-level incentive scheme. A practice visit by CCG AMS leads (or Prescribing Advisors) was recognised as a strategy used to conduct an antibiotic audit and offer feedback to practices. Additionally, local financial incentives were sometimes set up to encourage practices to conduct self-audit.

### 3.2. National Initiatives Contributing to QP Antibiotic Targets

Increased availability of prescribing data for CCGs and practices: Increased surveillance, availability and feedback of prescribing data following the introduction of QP were perceived to be associated with increased engagement with the QP scheme. This was also reported to be related to increased adoption of strategies like antibiotic audit and benchmarking.

National guideline on antibiotic prescribing: Participants also reported that having a national guideline helps in developing local strategies towards the QP antibiotic targets by providing a framework to underpin their AMS activities. In discussing the available guidelines on antibiotic prescribing, most of the participants indicated a preference for guidelines that were less frequently updated given the limited time allocated to AMS duties. In addition, most of the participants shared the view that the effect of the QP might be less about the financial incentive and more about having national targets and guidelines that promote antibiotic prudency. Even when their CCGs expected not to receive the QP financial payments (which depend also on other non-antibiotic targets), some participants reported that they still worked towards the QP antibiotic targets.

### 3.3. Challenges to Meeting QP Targets

One of the challenges to meeting the QP antibiotic targets reported by the participants was balancing the goal of AMS whilst ensuring access to antibiotics for those patients who needed them (dealing with sepsis was mentioned in this discussion by some participants).

Another reported challenge was engaging high antibiotic prescribers. AMS interventions such as practice visits, audits and feedback were often targeted by the CCGs specifically to high prescribing practices. Participants raised a concern about the possibility of disengagement by high prescribers due to continuously labelling and targeting them for most AMS interventions.

## 4. Discussion

Our paper summarises the strategies and activities reported by a selection of CCGs in England to facilitate antibiotic stewardship in primary care practices towards achieving the QP antibiotic targets. Evidence on the mechanism of impact of financial incentive schemes to improve the quality of care in primary care is insufficient given the limited number of rigorous studies on this [14]. We demonstrated possible pathways for the impact of QP on antibiotic prescribing in the conceptual model co-developed with CCG AMS leads who are also important users of AMS policy evaluation research.

Our findings are important to healthcare policymakers and quality improvement agencies in the planning and systemised implementation of programs towards antimicrobial stewardship. The participants reported some of the ways in which AMS strategies such as national guidelines, audits and feedbacks can be integrated to optimise their individual effects. The co-developed conceptual framework can inform a user-led investigation of the mechanism of the impact of financial incentive schemes on antibiotic prescribing in primary care practices. Given that the Quality Premium is one of the first national financial incentive schemes towards improvement in antibiotic prescribing in England, the developed framework depicting pathways for maximising the potentials of such a scheme can be useful in the development of future AMS financial incentive interventions.

The adoption of some of the AMS strategies identified in our workshop was also reported by a survey of CCGs in England [9]. Previous studies on AMS in outpatients have demonstrated the effectiveness of strategies like audit, prescribing feedback, guidelines, and prescriber AMS education and training in reducing antibiotic prescribing [15,16].

We recognise the limitation posed by the small sample size with regards to the representativeness of our findings. However, the diversity of the workshop participants (in relation to antibiotic prescribing behaviour and geographical location of their organisations) was important in capturing different perspectives and experiences, contributing to the credibility of our findings.

## Figures and Tables

**Figure 1 antibiotics-09-00044-f001:**
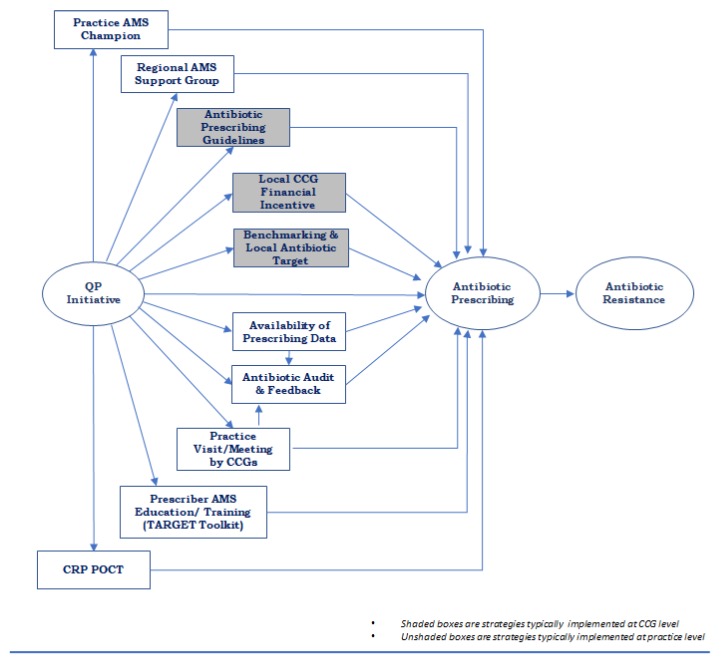
Conceptual model developed by participants showing pathways from Quality Premium (QP) to antibiotic prescribing and resistance.

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
