# Peer review of "Conceptualising the Integration of Strategies by Clinical Commissioning Groups in England towards the Antibiotic Prescribing Targets for the Quality Premium Financial Incentive Scheme: A Short Report"

_antibiotics, 2020, doi:10.3390/antibiotics9020044_

Round 1

Reviewer 1 Report

Thank you for the opportunity to review the manuscript by Philip Anyanwu et al.  The authors describe study based on a workshop of antimicrobial stewardship leads of Clinical Commissioning Groups (CCGs) from various regions of the English National Health Service, who are tasked with improving antimicrobial stewardship programs in their regions.  When regions achieve certain quality improvement goals, they are eligible to receive a financial incentive (the Quality Premium [QP]).  Antimicrobial stewardship in the outpatient setting is one of the elements included in the QP in 2015/16. The Workshop participants (10 CCG antimicrobial stewardship leads, 3 GPs and a nurse prescriber) broke up into small groups to discuss relevant innovations in their regions to achieve QP targets, then they altogether worked out a conceptual model of pathways from the QP to antibiotic prescribing, and finally, they undertook a discussion on how the QP had been implemented in their regions, capitalizing on the content of their earlier small group discussions.  Researchers summarized and took notes on all workshop discussions, and they performed a thematic analysis to identify main themes to develop the final conceptual model (shown in Figure 1).  The researchers identified 3 main themes from the workshop -- strategies implemented (practice visits, peer feedback, audits, local benchmarking, local incentives, often used in combination), contribution of national initiatives to achieving QP targets (improved surveillance and feedback of prescribing data, national guidelines), and challenges to meeting these targets (too frequent changes in guidelines, risk of disengagement by high prescribers). The researchers seek to illuminate the mechanisms underlying the impacts of financial incentives on meeting QP targets.  Interestingly, they found from the Workshop that financial incentives may not be major drivers in the implementation of QP targets.  

I did not find the conceptual model developed to be particularly clear or to have obvious practical utility given the Discussion provided. The text of the results read like a list of findings that might be better suited to presentation in a table.  

I have the following specific questions and comments:

1.  Can the authors expand on the utility of their findings and how they might be used in planning programs or establishing QP targets or guiding antimicrobial stewardship leads in their work?  Who is the targeted audience of this paper?  This can perhaps be part of an expanded Discussion section. The final sentence of the Discussion is difficult for me to understand; what is meant by "...inform a user-led investigation..."?  Who is a "user"?

2.  Do the findings of the workshop and the thematic analysis match findings of others in the literature on antimicrobial stewardship?  

3.  Can the authors add a table that summarizes the main Results, perhaps replacing much of the current text?

4.  Methods.  How were themes coded and then derived from the codes?  Was there a process used that involved agreement among more than one of the coders?  The Methods require more detail to be understood. 

5.  The sample of workshop participants is very small, and as it may not be representative of people in the sampled groups, this is a major limitation of the study that should be addressed. 

Author Response

Reviewer 1’s comment

Authors’ response

Can the authors expand on the utility of their findings and how they might be used in planning programs or establishing QP targets or guiding antimicrobial stewardship leads in their work?  Who is the targeted audience of this paper?  This can perhaps be part of an expanded Discussion section.

We have expanded discussions on the utility of our findings, further highlighting the targeted audience. (lines 154-163)

The final sentence of the Discussion is difficult for me to understand; what is meant by "...inform a user-led investigation..."?  Who is a "user"?

We have updated the manuscript to clarify that  “user” is in this context means CCG AMS leads. (line 152)

Do the findings of the workshop and the thematic analysis match findings of others in the literature on antimicrobial stewardship? 

Some of the antimicrobial stewardship strategies identified in our work have been reported in other studies. We have further expanded the discussion section to highlight how our findings fit those of previous studies. (line 164-167)

Can the authors add a table that summarizes the main Results, perhaps replacing much of the current text?

Given the qualitative nature and the need for a detailed description of the results, we do not think a table will make a substantial contribution to the presentation of the results. The conceptual model provides a visual representation of the findings from the workshop.

Methods.  How were themes coded and then derived from the codes?  Was there a process used that involved agreement among more than one of the coders?  The Methods require more detail to be understood.

The reviewer raised an important point on the need to expand the methods section. The manuscript has been updated with more details of the thematic analysis. (lines 77-82)

The sample of workshop participants is very small, and as it may not be representative of people in the sampled groups, this is a major limitation of the study that should be addressed.

We recognise the limitation posed by the small sample size with regards to representation of the CCGs in England, however, the diversity of the workshop participants (in relation to antibiotic prescribing rates and geographical location of their organizations) was important in capturing different perspectives and experiences. This has been included as a limitation in the manuscript. (lines 168-172)

Reviewer 2 Report

In the manuscript „Conceptualizing the integration of strategies by Clinical Commissioning Groups in England towards the antibiotic prescribing targets for the Quality Premium financial incentive scheme: a short report” of Philip Anyanwu and colleagues, the authors reported on outcome of a workshop exploring the experiences of antimicrobial stewardship (AMS) leads within Clinical Commissioning Groups in selecting and adopting strategies to help achieve the QP antibiotic targets.

Interestingly, the authors found that national targets, rather than financial incentives are the key for engaging stakeholders in quality improvement in antibiotic prescribing.

This reviewer feels that the information presented here are interesting and should be published. There are only two minor points that needs to be addressed in a revised form before the brief report is acceptable for publication.

Line 55: GP’s? What’s that?

Line 58: Was there any kind of people “selection” for the small working groups which might influence the outcome?

Author Response

Reviewer 2’s comment

Authors’ response

Line 55: GP’s? What’s that?

We thank the reviewer for pointing this out. We have included the full meaning with the abbreviation in parenthesis the first time it appeared intext. (line 55)

Line 58: Was there any kind of people “selection” for the small working groups which might influence the outcome?

In assigning participants to small working groups, we aimed to have diverse perspectives and experiences in each group in relation to regions and prescribing behaviour of the CCGs represented by the participants. This was important to facilitate comparison of experiences, driving creativity and inclusiveness in the development of the conceptual framework.

We have updated the manuscript to include the response above. (lines 59-63)

Round 2

Reviewer 1 Report

I have completed my review. My only comments are:

The authors have responded to all of my questions and comments appropriately. I have no further comments.